# Landslide Susceptibility Mapping: Machine and Ensemble Learning Based on Remote Sensing Big Data

**Bahareh Kalantar** [1],* , **Naonori Ueda** [1], **Vahideh Saeidi** [2], **Kourosh Ahmadi** [3], **Alfian Abdul Halin** [4] **and Farzin Shabani** [5]

1   RIKEN Center for Advanced Intelligence Project, Goal-Oriented Technology Research Group, Disaster Resilience Science Team, Tokyo 103-0027, Japan; naonori.ueda@riken.jp
2   Department of Mapping and Surveying, Darya Tarsim Consulting Engineers Co. Ltd., Tehran 15119-43943, Iran; saeidi@daryatarsim.com
3   Department of Forestry, Faculty of Natural Resources and Marine Sciences, Tarbiat Modares University, Tehran 15119-43943, Iran; kourosh.ahmadi@modares.ac.ir
4   Department. of Multimedia, Faculty of Computer Science and Information Technology, Universiti Putra Malaysia, Serdang, Selangor 45000, Malaysia; alfian@ieee.org
5   Global Ecology and ARC Centre of Excellence for Australian Biodiversity and Heritage, College of Science and Engineering, Flinders University, GPO Box 2100, Adelaide, SA 5001, Australia; farzin.shabani@flinders.edu.au
*   Correspondence: bahareh.kalantar@riken.jp; Tel.: +81-362-252-482

**Abstract:** Predicting landslide occurrences can be difficult. However, failure to do so can be catastrophic, causing unwanted tragedies such as property damage, community displacement, and human casualties. Research into landslide susceptibility mapping (LSM) attempts to alleviate such catastrophes through the identification of landslide prone areas. Computational modelling techniques have been successful in related disaster scenarios, which motivate this work to explore such modelling for LSM. In this research, the potential of supervised machine learning and ensemble learning is investigated. Firstly, the Flexible Discriminant Analysis (FDA) supervised learning algorithm is trained for LSM and compared against other algorithms that have been widely used for the same purpose, namely Generalized Logistic Models (GLM), Boosted Regression Trees (BRT or GBM), and Random Forest (RF). Next, an ensemble model consisting of all four algorithms is implemented to examine possible performance improvements. The dataset used to train and test all the algorithms consists of a landslide inventory map of 227 landslide locations. From these sources, 13 conditioning factors are extracted to be used in the models. Experimental evaluations are made based on True Skill Statistic (TSS), the Receiver Operation characteristic (ROC) curve and kappa index. The results show that the best TSS (0.6986), ROC (0.904) and kappa (0.6915) were obtained by the ensemble model. FDA on its own seems effective at modelling landslide susceptibility from multiple data sources, with performance comparable to GLM. However, it slightly underperforms when compared to GBM (BRT) and RF. RF seems most capable compared to GBM, GLM, and FDA, when dealing with all conditioning factors.

**Keywords:** big data; landslide susceptibility; flexible discriminant analysis; random forest; ensemble model

## 1. Introduction

The use of advanced remote sensing technologies has allowed high-dimensional multiresolution datasets to be conveniently accessible to researchers. Data of the earth's surface can be acquired from

satellite imagery that comes in various spectral, spatial, and temporal resolutions. Such images are not obtainable through unmanned aerial vehicle-mounted cameras. Such data are used to generate maps explaining the topography, land cover, lithology, etc., from which geospatial information can be extracted. This facilitates critical tasks and applications such as earth observation and management [1]. Nowadays, the sheer amount of remotely sensed data puts it into the category of big data. Sedona et al. [2] assert that remote sensing data needs robust processing and sense-making where (traditional) machine learning (ML) algorithms might be preferred over deep learning methods. This is because the latter necessitates massive amounts of training examples that involve millions of model parameters, whereas a properly formulated ML task can involve a lot less data. The work done by Shirzadi et al. [3] has also proven that, despite the deep learning trend, less complex ML can still work in a broad range of applications.

One critical application that has been on the rise, which has also benefitted from the availability of remote sensing big data, is landslide susceptibility mapping (LSM). LSM is a critical natural hazard mitigation strategy that accommodates for large scale destruction and losses [4]. There are several landslide predisposing factors that are grouped into human and environmental factors. These triggering factors are topographic, hydrological, lithological, land cover and manmade (e.g. construction and excavation) [3,5]. Data pertaining to terrain surface and human activities can be obtained via remote sensing methods, whose analytics are able to identify problems before the disaster strikes. Nowadays, more remotely sensed data can be acquired such as satellite imagery, aerial photogrametry, etc. [6], which can potentially make LSM more effective.

Recently, machine learning algorithms, together with strategies to apply them (either as a single classifier or via ensemble methods), have been extensively studied. Many ML algorithms are publicly available (written mostly in the Python and R languages), allowing models to be rapidly developed [3]. This trend has allowed remote sensing big data to be effectively used, analyzed and modelled. The main concern though is which algorithm to apply, and, when dealing with big data, which variables/features to select (dimensionality reduction) based on objectivity and bias [7,8].

*Related Studies*

LSM and the evaluation of landslide conditioning factors play a major role in landslide mitigation [8]. Ma et al. [9] categorized LSM models into inventory-based, knowledge-driven methods, data-driven methods and physically-based models. Pradhan et al. [10] further categorized data-driven methods into two models: (i) bivariate and (ii) multivariate (which is based on correlations among regional conditioning factors, as mentioned by Dou et al. [7]). Widely used bivariate models in LSM are Frequency Ratio [11,12], Weight of Evidence [10,13], Statistical Index [7,10] and Information Value [14]. Common multivariate statistical techniques include Logistic Regression [7,10,12,15–17] and Decision Trees [10]. Lately, machine learning (ML) algorithms such as Random Forest [13,18,19], Support Vector Machine [13,17,20], Artificial Neural Networks [11,17] and Naïve Bayes Classifiers [21,22] have also been used more in comparison to bivariate and multivariate models. ML algorithms have also been shown to be practical in LSM, which includes spatial prediction [23].

Supervised ML algorithms can be trained using data to create a model for the prediction of specific phenomena [24]. For example, Generalized Linear Models (GLM) and Generalized Additive Models (GAM) were reported by [13] and [25], respectively, where improved landslide susceptibility modelling was obtained by taking advantage of the linear and nonlinear relationships of the predictor variables. In other research [26], Boosted Regression Tree (BRT) and GLM were applied, along with Random Forest and CART (Classification and Regression Tree), to identify landslide prone areas. Their findings suggested that RF and GBM (BRT) performed well for LSM, and highlighted that GLM was also adequately competitive with other modern machine learning algorithms. Landslide conditioning factors, as well as a variety of conditioning factors, have shown to play significant roles in LSM. However, despite all the research, there is no standard guideline for factor selection to determine landslide prone zones [18]. This inspired researchers such as [18] to investigate the importance of factors through machine learning. Their results managed to decrease the input space down to only

12 conditioning factors (CF) for landslide occurrences in Shangnan County, China. The study in [4] also showed the benefit of reduced CFs. They finally used only 14 CFs to accurately identify landslide prone areas. They further discovered that when all factors were optimized and redundant variables removed, the accuracy of each model moderately improved. They noted, however, that their Random Forest classifier was insensitive to the presence of redundant data, signifying its potential robustness for LSM. Several other ML algorithms were also shown to be promising for LSM [8], namely SVM and K-means clustering. Interestingly, this work showed that landslide susceptibility prediction seems feasible with acceptable modeling efficiency, even when using unsupervised K-means. In addition, Flexible Discriminant Analysis (FDA) has shown promise for classification tasks other than LSM [27–29]. Solberg [30] asserts that FDA is a less computer-intensive algorithm to classify non-Gaussian features, requiring no advanced parameter specifications. For this reason, we aim to use FDA to map landslide prone zones and evaluate its results with other ML algorithms (i.e., GLM, GBM and RF) to verify FDA abilities in LSM application.

Besides training just one learning algorithm, an ensemble of algorithms (i.e., ensemble models) can be used to decrease noise and avoid overfitting [31]. Kordestani et al. [32] reported improved classification for flood susceptibility and groundwater potential mapping using ensemble models. Pham et al. [33] also employed an ensemble technique through a combination of Bagging, Dagging, AdaBoost, MultiBoost, Random SubSpace and Rotation Forest for landslide susceptibility mapping. They asserted that ensemble models show significant improvement over their nonensemble counterparts. Fang et al. [34] integrated the convolutional neural network (CNN) with three ML algorithms (i.e., SVM, RF, and LR) to identify landslide susceptibility zones in China. Their hybrid models were trained on 16 conditioning factors with an impressive 8.72% improvement in overall accuracy. For this reason, we are motivated to investigate ensemble models. Specifically, we will combine FDA, GLM, GBM (BRT) and RF to examine the degree of improvement in LSM.

In this study, 13 conditioning factors (altitude, slope, aspect, cross sectional curvature, profile curvature, plan curvature, longitudinal curvature, channel network base, convergence index, distance to fault, distance to river, valley depth, and lithology (geology) map) were prepared by analyzing and determining the most significant factors using variance-inflated factor (VIF). FDA was compared against GLM, GBM (BRT), and RF (some of the most extensively deployed models) in the R package. The tasks are (i) to create LSM, and (ii) to classify the maps into five categories indicating landslide probability proneness (i.e., very low, low, moderate, high, very high). Lastly, an ensemble model consisting of the four aforementioned models along with FDA, GLM, GBM (BRT) and RF was applied to the dataset, and all results were evaluated against statistical indices such as True Skill Statistic (TSS), Receiver Operation characteristic (ROC) curve, and kappa index.

Specifically, our research aims to answer four questions: (1) Is FDA appropriate when dealing with big data, multiscale datasets and landslide susceptibility mapping? (2) Is ensemble modeling weighted by the True Skill Statistics (TSS) criteria which are applicable for landslide susceptibility mapping in comparison with individual machine learning methods? (3) Which conditioning factors are the most/least influential for the identification of landslide susceptible zones in the study area? (4) How well does FDA perform against selected commonly used algorithm in LSM (i.e., RF, GBM (BRT), and GLM modeling)? The remainder of this paper is organized as follows. Section 2 outlines the geographical location of the study area and describes the inventory data and conditioning factors. Section 3 presents a generic overview of the methodological framework and detailed information about image processing, data derivation, FDA, GLM, GBM (BRT), and RF classification, and evaluation metrics. Section 4 describes the results, and Section 5 discusses the experimental findings. Section 6 provides the conclusions.

## 2. Study Area and Materials

### 2.1. Study Area

The Sajadrood catchment was selected as the study area, which is in the Mazandaran province of Iran. This location was selected as repeated landslides have occurred here [16]. As illustrated in Figure 2, Sajadrood has an area of 118.8 km$^2$ with a population of 26,809 (2006 census). It is situated between the north latitudes of 36°9′ and 36°10′ and east longitudes of 52°30′ and 52°40′. The land is mostly covered by dense forest towards higher altitudes, as well as orchards, agriculture, and paddy fields in lower altitudes in the north. According to the Iranian Meteorological Organization, temperatures vary from 2 °C in February to 38 °C in August, with humid weather. Throughout the year, Sajadrood experiences heavy rainfall, with an annual average of 680 mm. The geology of the area is predominately covered by sandstone, followed by silty marl, mudstone, limy marl and marly limestone, which are extended over the region [16]. Historical reports on landslide events reveal that rotational slides mostly happened in the area. For the most part, the landslide occurrence around villages and roads indicates that human activities are one of the triggering factors [4,16]. Additionally, the topography (altitude) and the geological characteristics of the study area induce more susceptibility to the hazard [16].

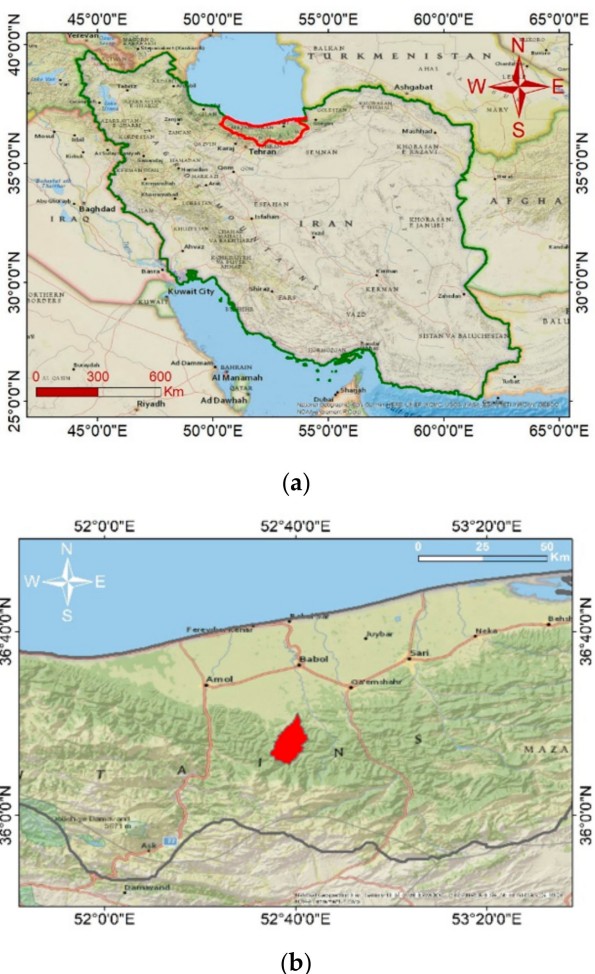

(a)

(b)

**Figure 1.** *Cont.*

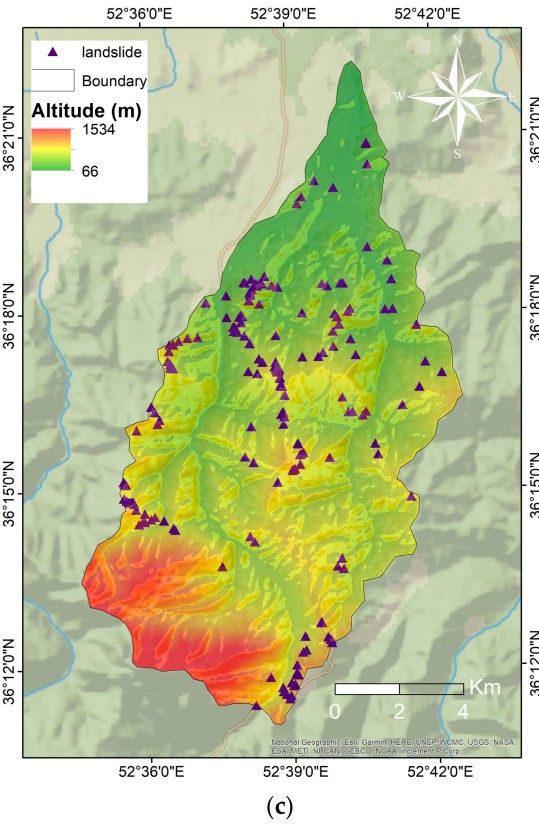

(**c**)

**Figure 2.** General location of the study area: (**a**) Iran; (**b**) location map; (**c**) digital elevation model (DEM) of the study sites including landslide inventory data.

## 2.2. Landslide Inventory Data Preparation

Similar to [18,35,36], field survey data, aerial photogrammetry, existing reports, and Landsat 8 imagery for the interpretation of historical events were used to prepare and update the landslide inventory map. In the study area, 227 landslides were identified as an inventory map. The inventory map is divided into the training and testing datasets. Specifically, 159 points (70% of the inventory locations) were allocated for training, whereas the remaining 68 points (30%) were used for testing. Landslide susceptibility mapping is a binary classification task where landslide indices are separated into the two classes: (i) landslide and (ii) nonlandslide. For this study, we randomly generated 10-sets for the nonlandslide class (227 points per set) using "create random point" in ArcGIS, and then we proceeded to divide each set from the landslide and nonlandslide classes (for a total of 454 points) for training and testing. At this stage, the selection of training and test points was completely random involving no human intervention. Defining landslide and nonlandslide pixels is a necessary part of the training process. The learning algorithm needs to receive data regarding both regions to develop the landslide model.

We applied the split-sample cross-validation approach (Figure 3), as per James et al. [37]. Basically, the initial dataset is randomly split into two partitions: the first (training) partition comprises 70% of the dataset, whereas the second (test) partition comprises 30%. A first split-sample iteration is made, the model parameters and evaluation metrics (assuming that several are measured, but it could also be only one) are stored, and a new split-sample iteration is made. The parameters and metrics from the second iteration are stored in the same place, and the iterations are repeated a total number of *R*-times. Hence, single iterations are similar to the single iterations in *k*-fold cross-validation, but the two approaches differ in terms of the number of runs that are made, i.e., only *k* in total for *k*-fold (unless the whole procedure itself is repeated multiple times), but *R*-iterations for repeated split-sample cross-validation (with $R \gg k$). Repeated split samples generate estimates of model parameters and evaluation metrics

which can be used to assess the model stability and assess uncertainty around the model parameters and evaluation metrics. In this sense, it is a more informative approach. One main limitation, however, is the computing cost, as running repeated models (e.g., 100) for large data can be compute intensive. With cross-validation, a more refined resampling procedure for split-sampling can be developed (e.g., using stratification) to reduce autocorrelation or to test the model's extrapolation ability.

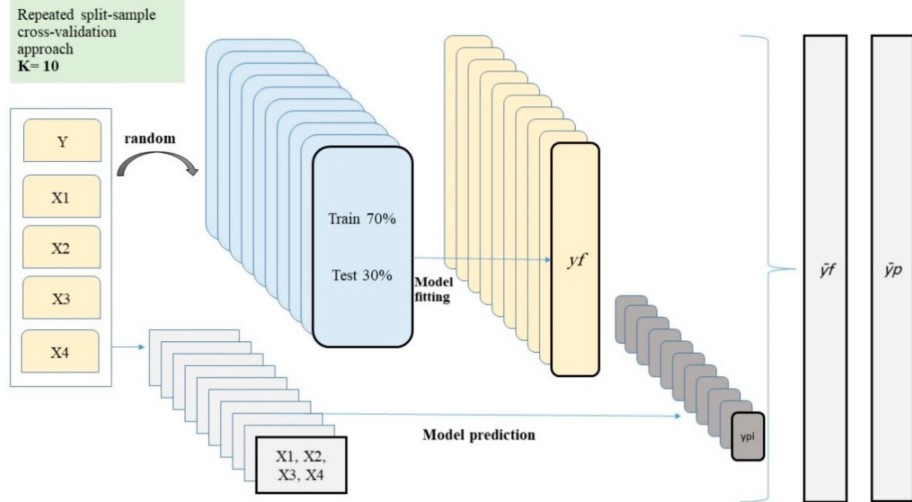

**Figure 3.** Procedure for split-sample cross-validation approach to evaluate a predictive model, illustrated for $k = 10$. $k$ is the fixed number of partitions, f represents the model function used to fit the landslide modeling, $yf$ is the vector of fitted values from the model fitted on all partitions except partition, and $pi$ is the vector of predictions made with model on partition $i$. $\bar{y}$f is the vector of mean $yf$ across all models (*for $i = 1$ to $k$*). $yp$ is the final vector of predictions made by appending the $k$ $ypi$ vectors, which can then be compared to the vector of initial observations y through the chosen evaluation metric.

### 2.3. Landslide Conditioning Factors

Based on existing and relevant literature, as well as data availability, we selected and classified 13 conditioning factors, namely lithology (geology) map, altitude, slope, aspect, cross sectional curvature, profile curvature, plan curvature, longitudinal curvature, channel network base, convergence index, distance to fault, distance to river, and valley depth. Our dataset has high dimensionality and comes from various sources with different spatial and temporal resolutions. To avoid complexity during modeling with such big data, we resampled the dataset to 10 m resolution. The selection and classification of the conditioning factors are explained as follows:

1.  *Lithology (Geology):* We used a 1:25,000-scale lithology (geology) map of the study area, which was obtained from the Geological Survey of Iran using satellite imagery. The dominant lithological units (e.g., sandstone and silty marl, mudstone, marly limestone and so on) in the area were classified into 9 classes, as per Figure 4a. Lithology (geology) was chosen, as it can be indicative of soil characteristics. These characteristics can be diverse and may influence erosion, ground stability and slide occurrence [17,35]. Table 1 shows the lithology (geology) of the Sajadrood catchment.

2.  *Elevation and Topographical Data:* We used a 1:25,000-scale topographic map of Sajarood to generate a 10 m Digital Elevation Model (DEM). The topographic map was provided by the National Cartographic Center of Iran from the aerial photogrametry. Since DEM derivatives can be utilized for geomorphological studies [38], first and second order DEM derivatives such as altitude, slope, aspect, cross sectional curvature, profile curvature, plan curvature, longitudinal curvature, convergence index, channel network base, and valley depth [39] can be very useful. We extracted them using SAGA GIS.

3.  *Altitude:* This CF can be very influential in landslide predictions [17,40]. In this work, altitude is derived from the study area's DEM, ranging from 66 meters (in the northern part) to 1534 meters (in the southern part) (Figure 4b).

4.  *Slope and Aspect:* Both these CFs were generated from the study area's DEM. For this study, the slope map was extracted to a maximum slope of 48° (Figure 4c). Aspect, which relates to meteorological and morphological characteristics, represents the horizontal direction of mountain slope faces [41]. The aspect map (Figure 4d) was divided into nine separate categories: (i) north, (ii) northeast, (iii) northwest, (iv) flat, (v) south, (vi) southeast, (vii) east, (viii) west and (ix) southwest. These two CFs were chosen, since slope directly influences the soil strength, and consequently, the landslide [42].

5.  *Curvatures:* Plan and profile curvatures are the descending flow acceleration (erosion/deposition rate) and the flow velocity variation of a slope, respectively [17]. Cross-sectional curvature, on the other hand, measures curvature perpendicular to the down slope direction to detect concave features such as channels (intersecting with the plan of slope normal and perpendicular to aspect direction). Longitudinal curvature calculates the curvature in the down slope direction (intersecting curvature with the plan of slope normal and aspect direction) [38,43]. For this research, plan curvature, profile curvature, cross-sectional curvature, and longitudinal curvature (Figure 4e–h) were manually classified into three categories: concave, flat, and convex shaped curvatures. More details and formulas regarding to curvatures are provided by Ehsani and Malekian [38], and Alkhasawneh et al [43].

6.  *The Convergence Index:* As another DEM derivative, the convergence index provides assessment of slope curvature (Figure 4i). This index describes the mean of the slope directions of neighboring pixels from the direction of the central pixel [39], which effectively indicates whether a pixel is convergent or divergent.

7.  *Channel Network Base and Valley depth:* Valleys and channels are considered as geomorphologic and hydrologic attributes [44]. The channel network base (Figure 4j) and valley depth (Figure 3) appear to influence landslides and debris flows distribution [45]. The channel network base uses elevation, flow direction, and divergence to calculate the network (http://www.saga-gis.org/saga_tool_doc/2.2.0/ta_channels_0.html). Valley depth contributes to drainage, that leads the way for the landslide. Therefore, it is based on the vertical distance to the depth contour lines (convergent) seen from the mountain ridges [44–46]. This can be estimated by subtracting the base level of the channel network from the DEM [47].

8.  *Distance to Fault and Distance to River*: From the topographic map, distance to fault and distance to river are generated based on the Euclidean distance function in ArcGIS (Figure 4l,m). These distances were chosen as landslides occurrence is most probable along the fault and river, due to erosion and ground instability [5,17,21].

**Table 1.** Characteristics of the Sajadrood Catchment.

| Symbol | Lithology (Geology) |
| --- | --- |
| Js | Shale with Intercalations of Conglomerate, Sandstone, Radiolarite, limestone and Volcanics |
| Mmsl | Marl, Calcareous Sandstone, Sandy Limestone, and minor Conglomerate |
| $K_2lm$ | Pale—Red Marl, Gypsiferous Marl and Limestone |
| $P_1c$ | Polymictic Conglomerate and Sandstone |
| Q | Low Level Piedment Fan and Vally Terrace Deposits |
| Plcm | Marl, Shale, Sandstone and Conglomerate |
| $TRe_2 = PeEm$ | Marl and Gypsiferous Marl Locally Gypsiferous Mudstone |
| Pesl | Sandstone, Calcareous Shale and Mudstone |
| $K_2lm$ | Hyporite Bearing Limestone |

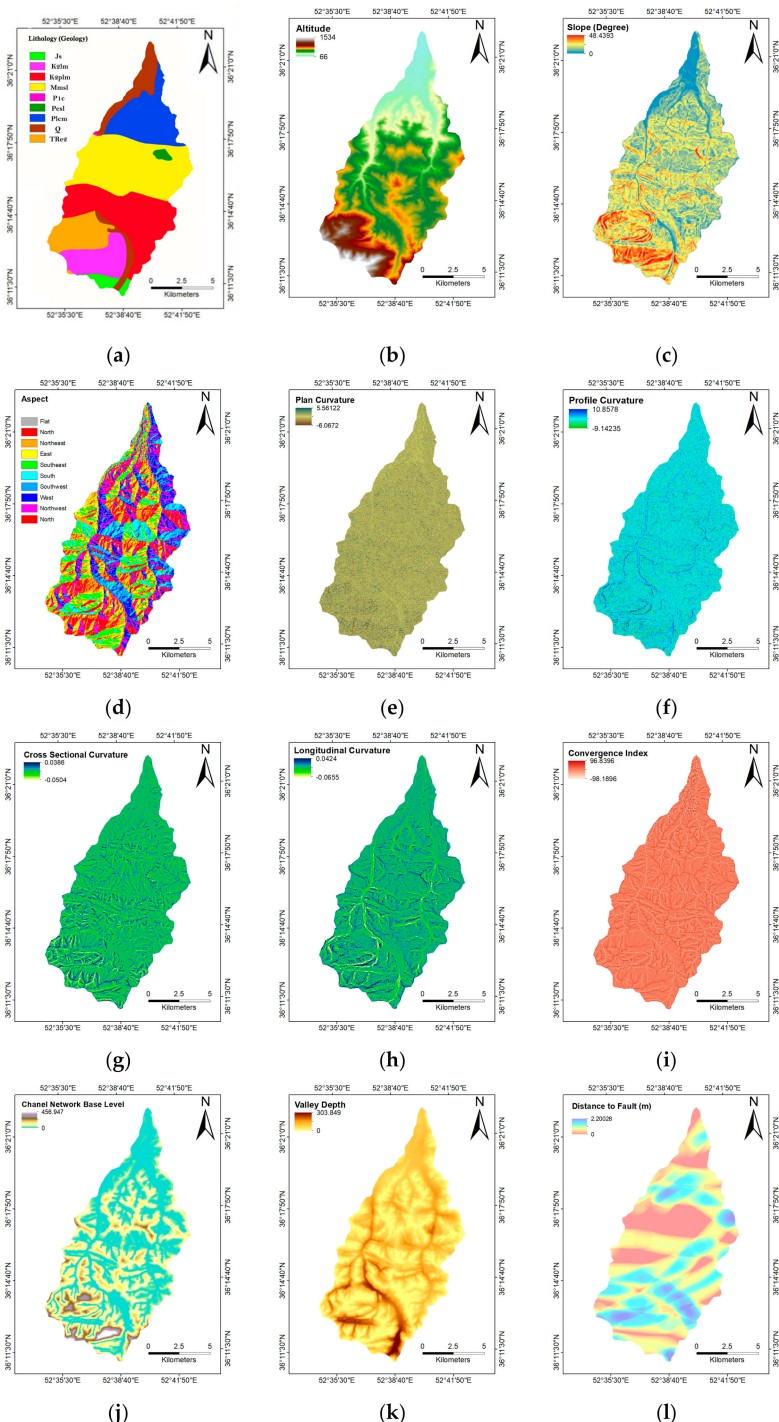

**Figure 4.** *Cont.*

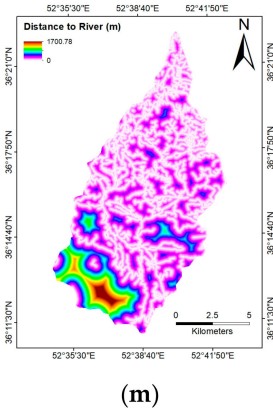

(**m**)

**Figure 4.** Landslide conditioning factors for LSM; (**a**) Lithology (Geology), (**b**) Altitude, (**c**) Slope, (**d**) Aspect, (**e**) Plan Curvature, (**f**) Profile Curvature, (**g**) Cross Sectional Curvature, (**h**) Longitudinal Curvature, (**i**) Convergence Index, (**j**) Channel Network Base, (**k**) Valley Depth, (**l**) Distance to Fault, (**m**) Distance to River.

## 3. Methodology

### 3.1. Overview

Figure 5 is a flowchart of the workflow in this work. The pixel values of the 13 conditioning factors were extracted into the landslide location points of ArcGIS 9.3, which was then imported into our R-language (version 3.0.2) for implementation. These served as training and test data for the principal and confirmatory models (FDA, GLM, GBM (BRT) and RF). Subsequently, the calculated coefficients of the landslide conditioning factors were converted into text format for the statistical Variance Inflation Factor analysis. Next, the four ML models and their ensembles were trained to map and classify landslide susceptible zones into five probability classes: (i) very low, (ii) low, (iii) moderate, (iv) high, and (v) very high. Finally, validations were carried out based on the metrics: (i) Receiver Operating Characteristics (ROC) curve, (ii) True Skill Statistics, and (iii) Cohen's kappa statistics.

### 3.2. Landslide Conditioning Factor Analysis

The conditioning factors analyses were done prior to modeling the machine learning algorithms [48]. Hence, both VIF and Pearson's coefficients were applied to identify any multicollinearity between the conditioning factors. As LSM deals with huge geospatial datasets, we decreased the dimensionality and multicollinearity by removing highly correlated factors using the USDM package version 1.1 [49].

#### 3.2.1. Variance Inflation Factor (VIF)

VIF is a common factor analysis method for landslide detection. It measures the degree of intercorrelation between the predictive variables [48] through the following Equation (1):

$$VIF = \frac{1}{1 - R'^2} \tag{1}$$

where $R\prime$ represent the multi correlation coefficient between an individual factor and other conditioning factors. In the current study, as per the standards of established previous works [50,51], VIFs greater than 5 or 10 indicate multicollinearity; therefore, that particular variable should be removed. Table 2 shows the factor analysis results via VIF. Seemingly, no values were greater than 5 for any conditioning factor.

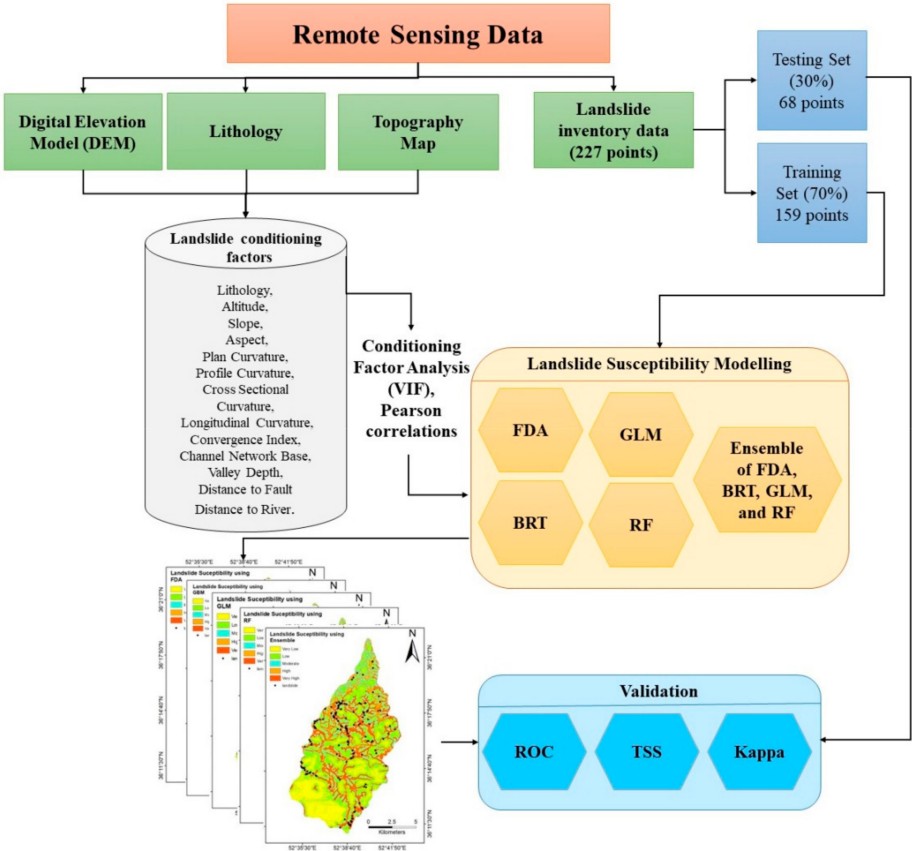

**Figure 5.** Framework of the proposed methodology.

**Table 2.** The Estimated Variance Information Factor (VIF) for Landslide Conditioning Factors.

| Conditioning Factors | VIF |
|---|---|
| Lithology | 2.26334 |
| Altitude | 1.321653 |
| Longitudinal Curvature | 1.531395 |
| Profile Curvature | 1.463843 |
| Plan Curvature | 1.463843 |
| Distance to River | 1.600335 |
| Slope | 1.391226 |
| Valley Depth | 1.695373 |
| Aspect | 1.024332 |
| Channel Network Base Level | 2.31 |
| Convergence Index | 1.834950 |
| Cross Sectional Curvature | 2.240171 |
| Distance to Fault | 2.514229 |

### 3.2.2. Pearson Correlations

Pearson's correlation coefficients (Equation (2)) is a measurement of correlation between two quantitative variables, or in our case, two conditioning factors. Highly correlated variables indicate linear dependence where they have the same effect on the response variable. Therefore, one of the variables can be removed from the model [50]. Independent variables with an *r* correlation value of

more than 0.70 should be removed [52]. In this work, the correlation between all independent variables yielded values less than 0.7.

$$r_{xy} = \sum_{i=1}^{n} \frac{X_i - \overline{X}}{\sum_{k=1}^{n} \left( X_i - \overline{X} \right)^2} \times \frac{Y_i - \overline{Y}}{\sum_{k=1}^{n} \left( Y_i - \overline{Y} \right)^2} \tag{2}$$

$X_i$ and $Y_i$ represent the respective value of $X$ and $Y$ for the $i$−th conditioning factor. $\overline{X}$ and $\overline{Y}$ are the mean of $X$ and $Y$, respectively. As mentioned, values greater than 0.7 indicate a high level of collinearity between variables/factors. The correlation between independent variables (Figure 6) showed that all variables were less than 0.7 and entered the model.

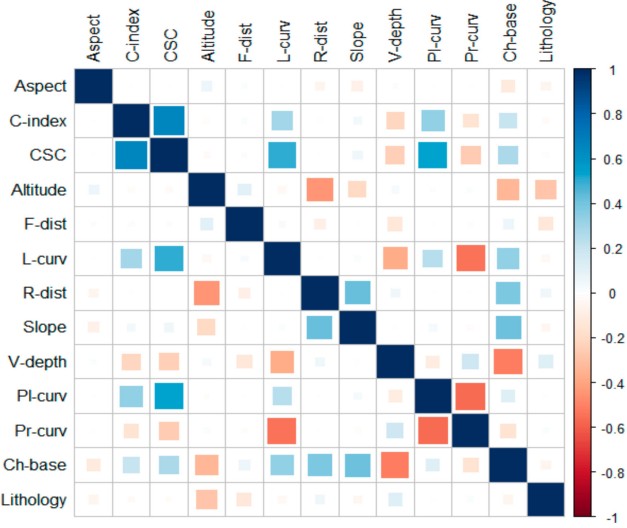

**Figure 6.** Pearson's correlations between landslide conditioning factors.

### 3.3. Machine Learning Algorithms

This section briefly addresses the supervised ML used for LSM, namely FDA, GLM, GBM (BRT), and RF, and ensemble models as well.

#### 3.3.1. Flexible Discriminant Analysis (FDA)

Flexible discriminant analysis (FDA) combines nonparametric regression models with nonlinear discriminant analysis, along with classification methods, into one framework [53]. FDA is more flexible for nonlinear classification tasks, since the clusters in FDA are smoother and softer due to the nonlinear transformation being applied [27]. FDA works well at classifying non-Gaussian features [30]. The FDA model in this research uses multivariate adaptive regression splines (MARS) which adaptively transform the predictors $x$ into the new space $h(x)$. The nonlinear discriminant analysis is then performed in the new space [53]. In this work, for a $J$ class of landslide events, the vector of canonical variates has a maximum of $K = J - 1$ components [27]. The fitted centroid for the $j$−th class in in this space (of the canonical variates) is calculated as:

$$\overline{\eta^j} = \sum_{g_i = 0} \frac{\eta(x)}{N_j} \tag{3}$$

Subsequently, the discrimination rule (weights) assigns an observation $x$ to class $j$, which minimizes the following objective function:

$$\delta(x, j) = \left\| D(\eta(x)) - \overline{\eta^j} \right\|^2 \tag{4}$$

where $D$ is the diagonal matrix of the sample class proportions that convert optimally scaled fits to discriminant analysis variables.

### 3.3.2. Generalized Linear Models (GLM)

The work in [13] demonstrated the effectiveness of Statistical Generalized Linear Models (GLM) for landslide susceptibility predictions. GLM is suitable for numeric variables because of its regression nature; however, it is sensitive to insignificant and correlated variables, which might degrade its performance [54]. GLM is an extension of the common regression structure for non-normal distributions, making it suitable for nonlinear and simple relationships between Gaussian and non-Gaussian distributed variables [41,54]. The model is explained by Equation (5).

$$Pr = \frac{e^{C_0 + C_1 X_1 ..... + C_n X_n}}{1 + e^{C_0 + C_1 X_1 ..... + C_n X_n}} \tag{5}$$

where Pr is the estimation probability of landslide occurrence, $C_0$ is the intercept, and $n$ is the number of independent variables. Terms $C_i$ and $X_i$ (for $i = 1\ to\ n$) represent the slope coefficients and the independent variables, respectively [23].

### 3.3.3. Generalized Boosted Regression Models (GBM) or Boosted Regression Trees (BRT)

Generalized Boosted Regression Models (GBM or BRT) combine statistics and machine learning to improve the performance of a single model (simple tree) for nonlinear classification tasks [28,42]. According to [32], GBM (BRT) prioritizes the importance of the conditioning factors during modeling. This makes it insensitive to outliers, and missing data can be modified to increase model accuracy through regression trees models and boosting algorithms [41,42]. The boosting step basically fits a sequential parameterized function to the gradient of the loss function in order to iteratively construct additive regression models [45]. Three hyperparameters exist, i.e., (i) shrinkage, which manages model complexity, (ii) learning rate, which controls the contribution of each tree to the entire model's construction, and (iii) bag fraction, which determines how many trees are needed to obtain the best fit [42]. The GBM (BRT) [32,55] algorithm is described as follows:

*Initializing weights of* $W_i = 1/n$,
*from* $m = 1$ *to iteration classifier of* $C_m$,
*Fitting* $C_m$ *to the weighted data,*
*Calculating misclassification rate of* $r_m$,
*Computing the classifier weight of* $\alpha_m log\left(\frac{1-r_m}{r_m}\right)$ *with* $\alpha_m$ *as the coefficient value,*
*Recalculating weights of* $w_i = w_i\ exp[\alpha_m I\ (y_i \neq C_m)]$

Finally, a majority vote is obtained by:

$$Sing = \left[\sum_{m-1}^{M} \alpha_m\ C_m(X)\right] \tag{6}$$

### 3.3.4. Random Forest (RF)

The Random Forest (RF) algorithm is a statistical technique with the ability to control a large number of highly correlated variables [13,18]. It is a decision tree-based, ensemble, parametric classifier that makes no assumptions regarding the data distribution, even when dealing with different types of input scales, variables, or even large datasets [56]. Li and Wang [54] and Chehata et al. [56] assert that RF is not sensitive to multicollinearity and redundant variables, and that RF measures the significance of variables and inputs as reliable indicators. Specifically, RF applies the Gini index to the significance of variables for its selections and decisions [57]. RF models also do not overfit the training data due to their independent random trees, and handle outliers, missing values, and noisy variables

effectively [41,56]. Therefore, we selected RF as one of our benchmarks to evaluate the performance of the newly applied FDA.

### 3.3.5. Ensemble of FDA, GBM (BRT), GLM, and RF

Selecting an algorithm from a set of available algorithms (for modeling) can be done based on certain evaluation criteria [58]. However, although passing some criteria, the model may not necessarily be the best for the forecasting task. Some models may be more sensitive than the model bias, which may also reduce the ability to transfer it [32,59]. One solution to avoid such problems is to use ensembles [60]. Ensemble modeling avoids selecting the single best model, which eliminates (or at least limits) model selection bias. Moreover, this kind of modeling provides a relative measure of the importance of each predictor among all candidate models [61].

In the present study, an analytical option for ensemble modeling and model averaging is to build predictions based on all (or a subset) of the models. This can be weighted by their weight of evidence or by their statistical performance. Researchers have proposed using predictive performance metrics that have been long used in modeling, such as the Area Under the Curve (AUC) of the ROC plot (AUC) [62] or the True Skill Statistics (TSS) [63], to weight the different models. In this study, TSS criteria were used to weight the models. These ensemble predictions can be obtained very simply, by calculating a weighted average of the predictions from all models. Models with a low weight of evidence basically have no predictive power, whereas models with similar weights will contribute similarly, allowing concurrent predictors to contribute equally. The weighted average prediction is thus expressed by the formula:

$$\overline{P} = \sum_{i=1}^{R} w_i P_i \tag{7}$$

where $w_i P_i$ is the weighted prediction (probability) from model *i*. The idea of the weight basically entails a weighted variance or standard deviation, and associated confidence intervals can also be easily estimated, providing a useful estimation of the uncertainty associated with the different candidate models. Finally, open source tool *R* was utilized to apply all models for this study.

### 3.4. Model Validation

To statistically evaluate the overall predictive accuracy of the models, True Skill Statistic (TSS), Receiver Operation characteristic (ROC) curve, and kappa index were calculated. The TSS metric performs a comparison between correct predictions, predictions from random guessing, to hypothetically perfect predictions. TSS can be calculated as:

$$TSS = sensitivity + specificity - 1 \tag{8}$$

Based on Equation (8), the resultant range of TSS is from −1 to +1. Perfect agreement is obtained when $TSS = +1$, whereas $TSS \leq 0$ indicates performance that is no better than random [63,64]. The area under the Receiver Operating Characteristic (ROC) curve, termed the AUC, is a quantitative measurement to evaluate predictive performance [10]. It can also be a representation of the success rate (prediction rate) and how well the model fits the data [42]. The curve is plotted with the x-axis representing the false positive rate (sensitivity) and the 100-specificity on the y-axis. The range for AUC is between 0 and 1. In landslide-based research, [11,65] state that $AUC \geq 0.7$ is considered acceptable.

Cohen's kappa statistic quantitatively measures the agreement between predicted and observed values, and reveals the degree of reliability of LSM [22]. Similar to AUC, kappa values also range from 0 to 1, where $Kappa = 1$ indicates the best model and 0 indicates otherwise [4].

## 4. Results

FDA, GLM, GBM (BRT), and RF, together with their ensemble models, were trained on the 13 landslide conditioning factors. Table 3 and Figure 7 show the weights and contribution of each factor during classification. Overall, *distance to river* made the largest contribution, followed by *lithology*, which was ranked as the second or third most important factor by all classifiers. The weights of other factors were less than 0.01, and varied from one algorithm to another. A weighting value of zero was obtained by the channel network base level in FDA, BRT, and GLM, while using RF method, the importance of this factor was 0.023 for LSM application. The RF model was more consistent for factor prioritization within all the conditioning factors.

**Table 3.** Factor Importance Using Four Machine Learning Algorithms.

| Conditioning Factors | FDA | GBM(BRT) | GLM | RF |
|---|---|---|---|---|
| Lithology | 0.03313 | 0.01437 | 0.03793 | 0.01674 |
| Altitude | 0.03125 | 0.00496 | 0.02332 | 0.0107 |
| Longitudinal Curvature | 0.01239 | 0.00954 | 0.0283 | 0.0058 |
| Profile Curvature | 0.0036 | 0.00546 | 0.00661 | 0.00457 |
| Plan Curvature | 0.00169 | 0.0042 | 0.00966 | 0.00451 |
| Distance to River | 0.83677 | 0.76437 | 0.79594 | 0.51263 |
| Slope | 0.01063 | 0.01501 | 0.02247 | 0.00693 |
| Valley Depth | 0.00213 | 0.00743 | 0.00456 | 0.00939 |
| Aspect | 0.00209 | 0.00375 | 0.00148 | 0.00435 |
| Channel Network Base Level | 0 | 0 | 0 | 0.02343 |
| Convergence Index | 0.00277 | 0.00334 | 0.00123 | 0.00226 |
| Cross Sectional Curvature | 0.00148 | 0.00192 | 0.00066 | 0.00235 |
| Distance to Fault | 0.1301 | 0.00572 | 0.00373 | 0.00528 |

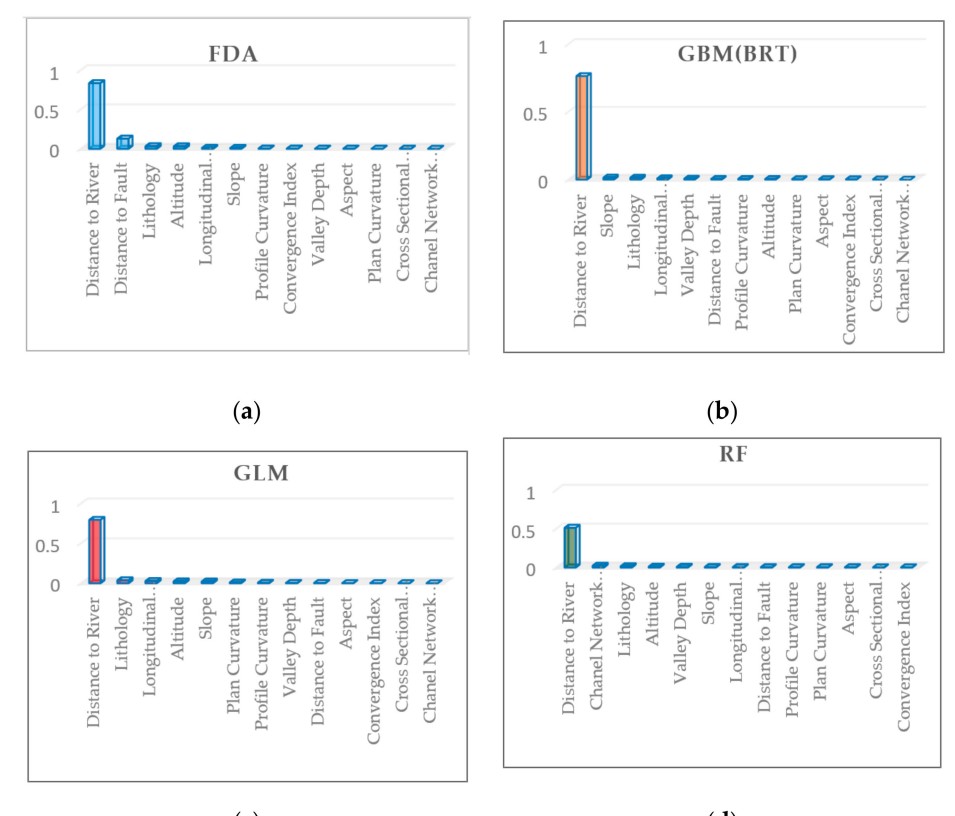

(a)

(b)

(c)

(d)

**Figure 7.** Factor importance for (**a**) FDA, (**b**) GBM (BRT), (**c**) GLM, and (**d**) RF (Bar plots are in descending order).

We evaluated the model's prediction capabilities using TSS, ROC, and Kappa. Table 4 shows the accuracies of the four models and their ensemble based on the 13 conditioning factors against validation indices. Each algorithm was run 10 times for sensitivity analyses, and the test errors were averaged. Accordingly, all models performed adequately, and the validation was satisfactory. Comparing the four ML models, RF had the best ROC, i.e., 0.8919 followed by BRT, at 0.8842. Though the differences between ROC between FDA and GLM were reported to be insignificant (at 0.8641 and 0.8604, respectively), in terms of TSS and the kappa index, FDA performed slightly better than GLM. Therefore, comparing the four models, the minimum TSS and kappa values belonged to GLM, while the maximum TSS and Kappa were obtained by RF. The best TSS (0.698), ROC (0.904), and kappa (0.691) belonged to the ensemble model.

**Table 4.** Validation Results for the 4 Models and Their Ensembles Using 13 Landslide Conditioning Factors.

| Dataset | FDA | | GBM (BRT) | | GLM | | RF | | Ensemble | |
|---|---|---|---|---|---|---|---|---|---|---|
| | avg | S.d | avg | S.d | avg | S.d | avg | S.d | avg | S.d |
| TSS | 0.614 | 0.0682 | 0.6551 | 0.0636 | 0.6096 | 0.0704 | 0.6869 | 0.0674 | 0.6986 | 0.0207 |
| ROC | 0.860 | 0.0343 | 0.8842 | 0.0309 | 0.8641 | 0.0357 | 0.8919 | 0.0300 | 0.9043 | 0.0148 |
| Kappa | 0.614 | 0.0688 | 0.6538 | 0.0641 | 0.6086 | 0.0710 | 0.6867 | 0.0676 | 0.6915 | 0.0204 |
| Sensitivity | 80.00 | 0.074 | 84.44 | 0.061 | 71.11 | 0.072 | 86.66 | 0.063 | 86.44 | 0.022 |
| Specificity | 82.22 | 0.065 | 80.00 | 0.053 | 84.44 | 0.073 | 84.44 | 0.061 | 84.22 | 0.024 |

Figure 8 shows the LSM classifications from the four models and their ensemble. The different colors represent the different probability risk classes, i.e., (i) very low, (ii) low, (iii) moderate, (iv) high, and (v) very high. The final landslide susceptibility maps show the regions that are classified into the five classes. Specifically, the values for each class are as follows: very low (<0.2), low (0.2–0.4), moderate (0.4–0.6), high (0.6–0.8), and very high (>0.8). All the models generated consistent general probability distributions in the five classes.

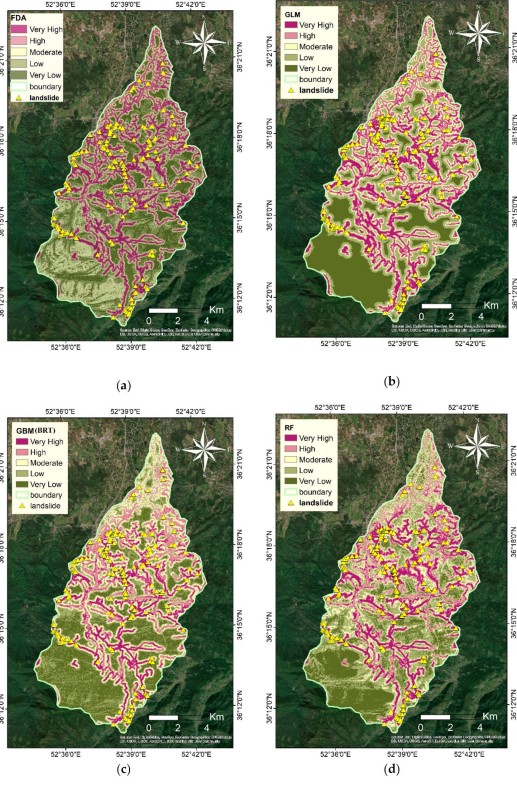

(a)

(b)

(c)

(d)

**Figure 8.** *Cont.*

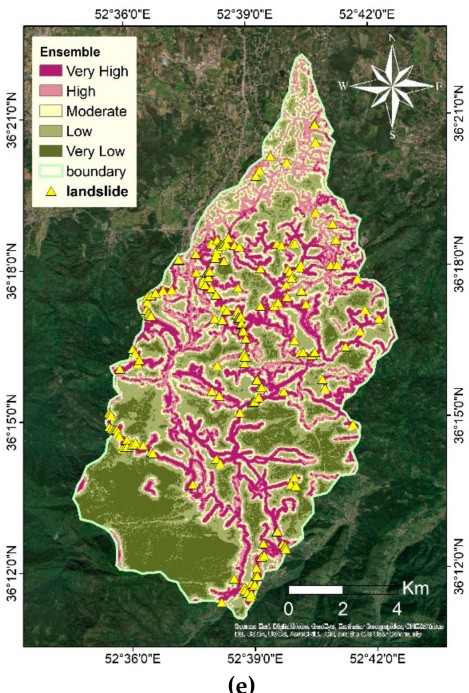

**(e)**

**Figure 8.** Landslide susceptibility maps by (**a**) FDA, (**b**) GLM, (**c**) GBM (BRT), (**d**) RF and (**e**) their ensemble models.

Table 5 shows the presence (coverage) of each probability class. Accordingly, all models classified most of the study area as very low risk for landslide events. More specifically, GLM, FDA, Ensemble, GBM (BRT), and RF mainly voted for very low probability with values of 56.1%, 59%, 64.4%, 69.51%, and 71.5%, respectively. FDA and GLM were more similar in voting percentages of the five classified probabilities within the region. While GBM (BRT) classified 15.06% of the area as a very high susceptible zone, other methods categorized it (very high zone) with a lower coverage (from 9.67% to 5.9%).

**Table 5.** The Ratio and Area of Each Susceptibility Class.

| Susceptibility | FDA | | GLM | | Models GBM (BRT) | | RF | | Ensemble | |
| --- | --- | --- | --- | --- | --- | --- | --- | --- | --- | --- |
| | % | Area (ha) | % | Area (ha) | % | Area (ha) | % | Area (ha) | % | Area (ha) |
| Very Low | 59 | 7408 | 56.14 | 7011 | 69.51 | 8680 | 71.51 | 8930 | 64.42 | 8044 |
| Low | 11.5 | 1436.5 | 12.19 | 1523 | 4.63 | 579 | 6.57 | 821 | 9.96 | 1244 |
| Moderate | 10.7 | 1343 | 11.71 | 1463 | 4.32 | 540 | 6.04 | 755 | 7.25 | 906 |
| High | 12.9 | 1616 | 12.5 | 1561 | 6.45 | 806 | 6.18 | 772 | 8.97 | 1121 |
| Very High | 5.9 | 683 | 7.43 | 928.5 | 15.06 | 1881.5 | 9.67 | 1208.5 | 9.38 | 1171.5 |

## 5. Discussion

In this study, the ML models and their ensemble were applied to remote sensing data to calculate and predict the probability of landslide events in the Sajadrood catchment. The extensive use of remote sensing techniques has made data and information from the ground surface and landslide conditioning factors (e.g., topography and geology), and their recent situations, available for our perusal. For example, the 30 m Landsat8-derived information regarding the current positions of landslides in the region helped us prepare the inventory map, which was the basis for training and validating the learning algorithms used in this study.

Using available datasets from the study area, 13 conditioning factors were resampled in the same resolution and then classified. Prior to model training, VIF and Pearson correlation factor analyses were applied to test the presence of multicollinearity in the datasets. The factor analyses step concluded that no high intercorrelation existed between the datasets, making it possible for no predictors to be removed from the dataset. Next, the training samples from the inventory maps and corresponding pixels of the conditioning factors were extracted and fed into the four-machine learning (ML) models and their ensemble. Finally, the probability maps were categorized into five classes.

The regression coefficients resulting from the four ML models suggest that the most important causal factor for landslides is distance to river, which shows a direct connection with landslide occurrence. This is consistent with several previous studies, i.e., [35,45,50]. Our regression results, however, indicate different weights for the conditioning factors within every algorithm; the major agreement about Channel Network Base Level importance was defined as being the least influential factor for LSM. Although the role of the Channel Network Base Level was not the same direction as it was in [45],, it reassured us that every conditioning factor plays a significant/insignificant role in specific topographies and study areas. The different and varied rankings among the models were the consequence of different regression procedures within the algorithms. Each of the ML algorithms used a special regression method, such as MARS (FDA), the linear regression (GLM), the Gini index in decision tree regression (RF), and the regression tree (BRT). Apparently, FDA, GLM, and BRT showed closer regression results (compared with RF) which can be interpreted as being comparable in nature in their modeling. RF, on the other hand, is free of any assumptions on the distribution of the variables, particularly with large inputs of different types and scales. At this point, the proposed FDA regression model demonstrated parallel performance with the GLM and BRT models (the two confirmed models in LSM) in the management of all types of conditioning factors. We observed that the lithology factor was the second most important conditioning factor in landslide phenomena; a similar pattern was presented by other researchers [26,35], emphasizing the importance and the role of lithology and soil types in landslide occurrence.

Validation of the four models showed that RF was the most reliable method, exhibiting the best prediction rate. The least certain algorithm was GLM. This underlines the robustness of RF and the sensitivity of GLM to the presence of correlated and redundant variables. Moreover, the results might also be due to the presence of both linear and nonlinear relationships between the various geospatial variables that RF could reliably consider. Our findings further indicate that the FDA model was adequate for LSM, as it was compatible with RF, GBM (BRT), and slightly better than GLM, as shown in Figure 9. The similar pattern, distribution, and percentages of each susceptibility class in the proposed FDA, compared with the commonly used GLM in landslide prediction, seemingly testifies to the applicability of FDA for LSM (Table 5). Figure 9 presents the accuracy and reliability of the four ML models based on ROC and kappa indexes. The GLM model created district boundaries between the probability classes and did not properly represent natural phenomena such as landslides, which is in agreement with [54]. The authors of that paper state that the presence of correlated variables decreases the precision and certainty of GLM. A close comparison between FDA and BRT revealed that FDA does not require the setting of parameters prior to modeling, whereas BRT necessities trial and error for (some) parameter specifications. For this reason, the use of FDA is more straightforward, especially when dealing with remotely sensed big data. Regarding four models and their ensemble, the ensemble



model outperformed others. This suggests that accuracy improvements can be achieved using an ensemble model. With respect to the fusion and combination concept in both the ensemble and hybrid models, our finding was in agreement with [34], where the integration of the models could lead to better performance for LSM. Evidently, the higher efficiency and performance are the result of the ensemble model. It considerably improves the accuracy and certainty of the predictions by suppressing the weaknesses and disadvantages of each individual model, and by taking advantage of the responses of the combined models [17,66,67].

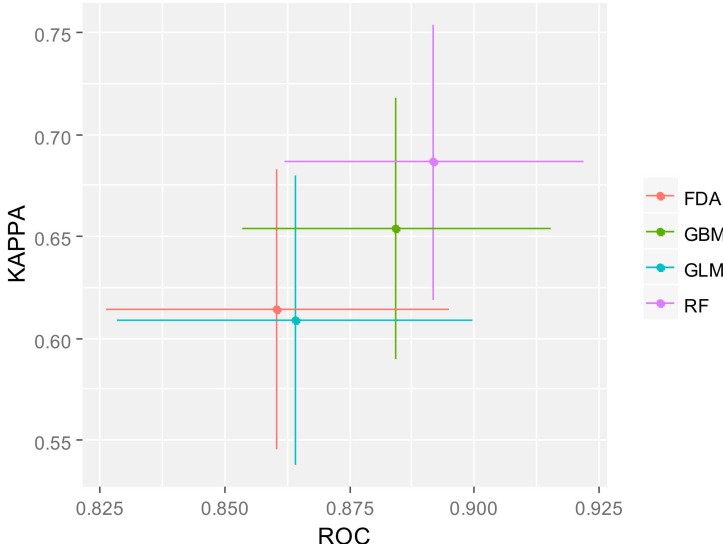

**Figure 9.** ROC and Kappa tests for ML algorithms.

The classification results illustrate low and very low probabilities of landslide occurrence in the southern parts of the study area. These parts are higher in altitude, more densely forested and more characterized by limestone, and have minor streams and rivers. In this context, the training seed might be affected by specific conditions in that part of the study area, which might be the result of inaccurate landslide inventory maps at the highest altitudes, inaccessible places (in terms of field surveys and reports) and dense forests (in cases of aerial photogrammetry). Another reason for this result could be the presence of root of vegetation (forest land cover). By looking at the LSM results and the distribution of landslides throughout the study area, we found that distance to river was a major landslide inducing factor.

## 6. Conclusions

To mitigate the tragedy, property losses, and human casualties caused by landslides, precise landslide susceptibility mapping (LSM) is necessary. This study provides insights into LSM modeling based on the FDA algorithm, which, in turn, is based on 13 conditioning factors. The factor analysis results emphasize the influence of river erosion on landslides. Therefore, risk mitigation recommendations can be enforced, such as soil strengthening (through roots vegetation, bricks, stones, blocks or concrete), defining a safe distance for construction projects (i.e., to be further from rivers), and landslide maintenance easement (to decrease the socio-economical loss). LSM in this context can therefore be used as a guideline for these recommendations, where it could be useful for decision makers.

As per conditioning factors analyses and their importance, simultaneously using different regression algorithms seem to be more reliable, as the majority votes could decide on which factors were significant in the region. Regarding the algorithmic modeling part, our findings suggest that the ensemble model was more robustly effective than the single algorithmic models, i.e., RF, GBM (BRT), FDA, and GLM. The single models also required more tweaking. For instance, redundant data had to be removed in GLM to decrease sensitivity to the presence of correlated variables. This involved

multicollinearity analyses of the datasets to assess the significance of the variables and reduce large quantity data before modeling. The RF algorithm was better able to handle noisy data and correlated variables. The application of the FDA algorithm to LSM was a success, and its ability to manage enormous datasets seems viable. Consequently, we intend to perform further investigations using FDA for LSM by applying other conditioning factors (e.g., distance to road, land use, etc.). Also, we intend to include factor optimization in other study areas with different terrains and geological characteristics. In future works, we plan to improve the accuracy and certainty of the FDA model and develop hybrid models, since the latter (e.g., ML algorithm integration with a convolutional neural network) seem to give good results. Further investigations will be on the basis of other ensemble techniques and their combinations with FDA to fully understand the advantages and limitations of this technique.

**Author Contributions:** B.K. and K.A. acquired the data; B.K. and K.A. conceptualized and performed the analysis; B.K. and V.S. wrote the manuscript, discussion and analyzed the data; N.U. supervised and the funding acquisition; B.K., A.A.H. and F.S., provided technical sights, as well as edited, restructured, and professionally optimized the manuscript. All authors have read and agreed to the published version of the manuscript.

**Funding:** The APC is supported by the RIKEN Centre for Advanced Intelligence Project (AIP), Tokyo, Japan.

**Acknowledgments:** The authors would like to thank the RIKEN AIP, Japan for providing all facilities during the research.

**Conflicts of Interest:** The authors declare no conflict of interest.

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
