# Peer review of "Landslide Susceptibility Mapping: Machine and Ensemble Learning Based on Remote Sensing Big Data"

_remotesensing, doi:10.3390/rs12111737_

Round 1

Reviewer 2 Report

The paper presents a comparison between different data-driven models for the assessment of landslides susceptibility in a catchment of northern Iran. An ensemble model was also created and tested to verify its predictive capability compared with the other models.

The paper is interesting, well written and well explained in all its sections. Instead, I do not see how the use of remotely-sensed data allowed to improve this research. Since this paper is submitted to Remote Sensing journal, I suggest the Authors to highlight well, in the Discussion section, the role of remotely-sensed data in this research, their utility, their significance in improving the analyses and the results.

Other suggested revisions follow:

  • There are no lines numbers. Authors have to add these to improve the comprehension in the review phase
  • Several references in the text are presented in wrong way respect to the rules of the journal. Please correct them
  • Landslide susceptibility is not a hazard mitigation strategy. It could be the preliminary task for hazard and risk assessment and could be used for land planning. Please, clarify this aspect
  • “Several human and environmental factors can cause landslides such as topographic, hydrological, lithological, land cover and manmade factors”: it is better adfirming that several factors predispose landslides triggering.
  • Landslides susceptibility models have to be classified in knowledge-driven, data-driven and physically-based methods. Bivariate and multivariate methods are two types of data-driven techniques
  • “They discovered that slope aspect, altitude, distance to faults, lithology, normalized difference vegetation index, plan curvature, profile curvature, distance to rivers, distance to roads, slope angle, stream power index, and topographic wetness index (12 factors) were important trigger factors for landslide occurrences”: this is not valid anywhere, but only in that context. Please clarify it
  • “3) Which triggering factors are the most/least influential for identification of landslide susceptible zones in the study area?”: A bit of confusion, in landslides susceptibility models there are predisposing (predictors) factors, not triggering (e.g. rainfall). Please clarify this and correct throughout the paper this error
  • Please, provide also a geological map of the study area
  • Add a detailed description of the landslides generally occurred in the study area. What type of landslides are? Have similar classification or not? Are deep or shallow? What are the main triggering conditions? Triggering events?
  • Justify the choice of 70:30 ratio between training and test datasets
  • Indicate the range of probability of each susceptibility class and the methodology used to discriminate these ranges
  • Authors should calculate also true positives, true negatives, false positive and false negatives as indexes for the evaluation of the models
  • A more detailed discussion about the utility and the effectiveness of the ensemble model is needed

Round 2

Reviewer 1 Report

The manuscript has been considerably improved and I suggest to accept it for publication after minor revisions.

Author Response

Dear Reviewer,

We would like to thank reviewers for their comments. We improved the English language and have checked the spell. Moreover, we did a minor revision on the presentation of results and methods.

Best regards,

Bahareh Kalantar

Reviewer 2 Report

The paper was improved significantly in its revised version. I suggest the acceptance of the paper in this version.

Author Response

Dear Reviewer,

We would like to thank for the acceptance of the paper in current version. However, we improved the English language and have checked the spell. Moreover, we did a minor revision on the presentation of results and methods.

Best regards,

Bahareh Kalantar